# First Report of *Clonostachys rosea* as a Mycoparasite on *Sclerotinia sclerotiorum* Causing Head Rot of Cabbage in India

**DOI:** 10.3390/plants12010199

**Published:** 2023-01-03

**Authors:** Ruppavalli M. Venkatesan, Karthikeyan Muthusamy, Johnson Iruthayasamy, Balakrishnan Prithiviraj, Parthiban V. Kumaresan, Pugalendhi Lakshmanan, Irene Vethamoni Perianadar

**Affiliations:** 1Department of Plant Pathology, Tamil Nadu Agricultural University, Coimbatore 641003, India; 2Department of Plant, Food and Environmental Sciences, Faculty of Agriculture, Dalhousie University, Truro, NS B2N 5E3, Canada; 3Department of Vegetable Sciences, Horticultural College and Research Institute, Tamil Nadu Agricultural University, Coimbatore 641003, Tamil Nadu, India

**Keywords:** biological control, mycoparasitic fungus, sclerotia, SEM, sporodochia

## Abstract

*Clonostachys rosea*, an ascomycetous, omnipresent, cellulose-decaying soil fungus, has been reported to be a well-known mycoparasitic biological control agent. In this study, we isolated *C. rosea*, a mycoparasitic fungus for the first time in India from sclerotia of the notorious plant pathogen *Sclerotinia sclerotiorum*, causing head rot disease in cabbage. A total of five mycoparasitic fungi were isolated from the sclerotial bodies of *S. sclerotiorum* (TNAU-CR 01, 02, 03, 04 and 05). All the isolates were tested under morpho-molecular characterization. Among them, TNAU-CR 02 showed the greatest mycelial inhibition of 79.63% over the control. Similarly, the SEM imaging of effective *C. rosea* isolates indicated the presence of numerous conidia destroying the outer cortex layers of sclerotia. Metabolite fingerprinting of *C. rosea* TNAU-CR 02 identified 18 chemical compounds using GC-MS analysis. The crude antibiotics of *C. rosea* TNAU-CR 02 were verified for their antifungal activity against *S. sclerotiorum* and the results revealed 97.17% mycelial inhibition compared with the control. Similarly, foliar application of TNAU-CR 02 at 5 mL/litre on 30, 45 and 60 days after transplanting showed the lowest disease incidence of 15.1 PDI compared to the control. This discovery expands our understanding of the biology and the dissemination of *C. rosea*, providing a way for the exploitation of *C. rosea* against cabbage head rot pathogens.

## 1. Introduction

*Sclerotinia sclerotiorum de Bary* is a dreadful pathogen that lives in soil and has a diverse host range, worldwide. It infects more than 400 plant varieties and causes massive economic losses [1]. The pathogen is able to survive in soil for long periods of time as a type of resting structure called a sclerotium [2,3]. Because of its hard melanized rind it can withstand chemically and physically adverse environmental conditions [4]. It can germinate either myceliogenically or carpogenically, producing windborne ascospores that infect aboveground crop parts [5]. Though the pathogen germinates in two ways, myceliogenous germination is the key mode of pathogenic infection which can lead to huge financial losses. Disease outbreaks occur in drier areas during the summer months where irrigation provides favourable climate conditions for disease development [6,7]. To manage the disease, the unremitting application of fungicides leads to development of resistance among the pathogen, *S. sclerotiorum*. Soil application of fungicides may cause microbial community imbalance in the soil and suppress beneficial soil microflora [8]. While the pathogen is extremely challenging to manage because of its survivability in the resting structure, a variety of management methods have been utilized to decrease their survival and abundance [9].

In general, Sclerotia are infested and destroyed in the field by a variety of mycoparasites, such as *Sporidesmium sclerotivorum*, *Coniothyrium minitans*, *Clonostachys rosea*, *Talaromyces flavus*, *Trichoderma harzianum*, etc., These parasitic fungi attack the surviving structures, pierce and invade the hard outer layer of the sclerotia, colonising the internal cells, and taking nutrients from them, causing them to die [10]. One of our primary goals was to identify the fungal strain that is selectively responsible for mycoparasitising and targets the pathogen’s resting structure.

Schroers, Samuels, Seifert and Gams, previously identified *Gliocladium roseum*, belonging to the family Bionectriaceae [11], as an omnipresent, cellulose-decaying soil fungus commonly found in an unusual array of environmental situations in tropical and temperate regions, and even in sub-arctic and desert regions of the biosphere [12]. Although, it is omnipresent in nature, most isolates or strains fitting this biocontrol agent have been reported from South American, the Netherlands, China, Mexico, Poland, Denmark, Canada, Brazil, etc., [11,13,14,15,16]. It has been repeatedly reported in cultivated higher plant rhizospheres, grasslands, forests and coastal soils, nematodes, freshwater, and so on [17]. Despite being recorded in many different places, *C. rosea* has primarily been isolated and studied from the soil ecology and distinct plant parts. The endophytic nature of *C. rosea* has also been reported in various tissues, colonizing plant stems, roots, seeds of soybean, onions, red clovers, and the leaves of strawberry and raspberry plants [18,19].

Earlier reports have shown that *C. rosea* has a greater destructive mycoparasitic activity in various plant pathogenic fungi and a superior antagonistic activity in managing various plant diseases, thus promoting plant growth [15,20]. This has been accompanied by the biocontrol of sclerotia of *Sclerotinia sclerotiorum* of soybean on a regular basis [21,22,23,24,25,26], cysts of *Heterodera* sp., *Verticillium dahliae* Kleb. [27], several species of *Botrytis* [28], *Globodera* spp. and other soil fungi and plant materials [17]. Although this biocontrol agent has been extensively studied in soybean and pea plants infected with *S. sclerotiorum*, there are no published articles on this fungus’s use as a biocontrol agent against *Sclerotinia scleorotiorum*, in India where it mycoparasitizes the sclerotia of cabbage head rot pathogens. It affects all growth stages of the crop and causes over million dollars causing commercial yield losses in field, storage and marketing conditions [29].

The objective of our study was (1) to explore the mycoparasitic fungal isolates, *C. rosea* associated with the sclerotia of *S. sclerotiorum* infecting cabbage crops; (2) to assess the mode of action of *C. rosea* against cabbage head rot pathogens; (3) assess the antifungal organic compounds produced by the effective mycoparasite against *S. sclerotiorum* through gas chromatography–mass spectrometry (GC-MS) and; (4) to assess the liquid formulation of effective *C. rosea* for the management of cabbage head rot pathogens under greenhouse conditions.

## 2. Results

### 2.1. Isolation of Mycoparasitic Fungi from the Infected Sclerotia of Cabbage

A total of five fungal isolates were isolated from mycoparasitized cabbage head rot-infected sclerotia. The collected scleortia showed a slight yellowish pigmented mycelial growth on the outer sclerotial surface. Those presumed mycoparasite-sterilized sclerotia were placed in PDA medium supplemented with rifampicin. Twenty days after inoculation, mycoparasitic fungi were isolated from the infected sclerotia, wherein the fungus initially produced white to yellowish mycelial growth with the presence of orange-coloured irregular to round-shaped sporodochia (Appendix A). Pure culture of the fungus was isolated by separating the orange sporodochia produced during isolation and placing them on fresh PDA media supplemented with rifampicin (Figure 1, Appendix A). After 15 days of inoculation, microscopic examination of the fungal mycelium from the inoculated sporodochia revealed the presence of multiple branched septate mycelia to construct hyaline conidiophores with smooth, lobed to round conidia (Appendix A). Based on the above-mentioned morphological characteristics, the fungi were identified as *Clonostachys* sp., and all five isolates were labelled as TNAU-CR 01, TNAU-CR 02, TNAU-CR 03, TNAU-CR 04 and TNAU-CR 05, respectively.

### 2.2. Molecular Characterization of Clonostachys sp. Isolates

The genomic DNA isolated from five *Clonostachys* sp. was subjected to PCR amplification using primers corresponding to the 5.8S rRNA region. The DNA from all five isolates was amplified using a PCR amplification size of approximately 580 bp. The rDNA homology sequence alignment with BLAST indicated that the nucleotide sequence identity was more than 98% identical with existing isolates of *C. rosea* available in the NCBI database. Nucleotide sequences from the ITS region of all the *Clonostachys* sp. isolates were submitted to NCBI and designated their accession numbers (MZ754407, ON025052, ON025053, ON02505 and ON025055). A phylogenetic tree was also constructed using the neighbour-joining method, which revealed the formation of three different clades, indicating each isolate phylogenetically varied from each other. Close *Clonostachys* species exhibiting different life forms were analysed by sequence similarity from the five isolate strains, showing all five isolates were significantly similar. The sub clade 1 was phylogenetically 77% similar with our *C. rosea* isolates. In major sub clade 2, isolates TNAU-CR 01, TNAU-CR 02 and TNAU-CR 04 were 80% similar with each other. *Trichoderma asperellum* isolate *PKU F7* (KC113288.1) was kept as an outgroup (Figure 2). The phylogenetic analysis revealed that *C. rosea* TNAU-CR 01 and TNAU-CR 02 isolates were phylogenetically identical, while the other *C. rosea* isolates differed from each other.

### 2.3. Dual Culture Assays

A co-inoculation method was performed to test for antagonism. A 9 mm mycelial disc of *S. sclerotiorum* and *C. rosea* was simultaneously incubated on PDA dishes where both colonies grew towards one another without any evidence of inhibition zones between the pathogen and the biocontrol agent, even after 10 days of incubation. Subsequently, a pre-inoculation study was performed where a 20-day-old mycoparasitic fungal culture was inoculated 7 days before pathogen inoculation. Here, hyperparasitism and inhibition zones were observed in abundance at the interaction site 20 days after inoculation. After the interaction, the mycelium started infecting and covering the sclerotia, forming small yellow colonies over the sclerotia, clearly revealing mycoparasitic activity. Later, mycelium of *C. rosea* started infecting and covering the sclerotial bodies of the pathogen. Where the sclerotia had become colonized by the mycelium on the PDA medium, the sclerotia had become ruptured, killed and distorted. Though *C. rosea* was slow in mycelial development it quickly proliferated and invaded inside the hard cortex of the sclerotial rind producing a great number of conidiospores. The daily growth rate (DGR) scope of the strain TNAU-CR 02 was 2.9–4.0 mm per day. It was characterized that among the five *C. rosea* isolates, TNAU-CR 02 showed the highest mycelial inhibition at 79.63% over the control. This was followed by TNAU-CR 03 which recorded an inhibitory impact of 74.07% over the control (Figure 3, Table 1). As a result of our findings, a more rapidly developing and faster mycoparasitising strain may play a significant role in the mycoparasitisation of target microorganisms. Thus, we focussed on the *C. rosea* TNAU-CR 02 isolate in subsequent experiments.

### 2.4. Identification of the Mycoparasitic Growth of C. rosea against S. sclerotiorum Using Scanning Electron Microscope

Characteristic symptoms of mycoparasitism by *C. rosea* TNAU-CR 02 include profuse raised fungal growth with white to yellowish mycelial growth on the outer surface of the sclerotia followed by mycoparasitized hyphae which invading both the exterior and interior surface of the sclerotia; subsequently, the sclerotia become devastated, distorted, softened, and collapse. The visible light orange-coloured fungal fruiting bodies, called sporodochia, were observed after subculturing the growth on PDA. While disturbing the sporodochia structure, the biocontrol agent forcibly released a huge mass of hyaline, round conidiospores.

Scanning electron microscopic sections of the 20-day-old *C. rosea*-infected sclerotial im confirmed the incidence of mycoparasitism. The single sporodochial size of *C. rosea* TNAU-CR 02 was 103.06–146.5 µm in width and 149.9–194.3 µm in length (Figure 4). The conidiospores remained smooth, lobed to round 1.585–4.445 µm in length. The whole conidiophore was round in shape with a size ranging from 11.95–19.23 µm in length (Figure 5). The *C. rosea* TNAU-CR 02-infected sclerotia showed a shrunken, distorted and shrivelled appearance with numerous conidiospores surrounding the outer and inner layers of the sclerotia [26]. When cut open, the infected inner surface of the sclerotia showed decay and a distorted appearance. The development of appressoria-like spore attachment structures was seen at the point of contact in the cortex layer of the sclerotia. As the biocontrol agent produced numerous conidiospores, the outer cortex was finally damaged and the whole sclerotia was destroyed.

### 2.5. Metabolite Fingerprinting of the Volatile/Non-Volatile Bioactive Compounds Extracted from the Ethyl Acetate Fractions of C. rosea TNAU-CR 02

The chemical compounds present in the ethyl acetate fractions of *C. rosea* TNAU-CR 02 showed a total of eighteen bioactive metabolites. These included: dihydro-3-(2H)-thiophenone; dl-glyceraldehyde dimer; propanedioic acid, phenyl-; butanamide, 2-hydroxy-N,2,3,3-tetramethyl; à-methyl 4-O-methyl-D-mannoside; 4-methyl(trimethylene)silyloxyoctane; 1-tetradecanamine, N,N-dimethyl-; pilocarpine; hexadecanoic acid, 2-hydroxy-1-(hydroxymethyl)ethyl ester; pentadecanoic acid, 13-methyl-, methyl ester; 9-hexadecenoic acid; octadecane, 3-ethyl-5-(2-ethylbutyl)-; erucic acid; 8,11-octadecadienoic acid, methyl ester; 9-octadecenoic acid, methyl ester, (E)-; 13-docosenamide, (Z)-; cis-11-eicosenamide and 17-pentatriacontene. A peak identification was carried out by comparing the mass spectra and retention times (*t*R) against reference standards, the NIST database, and literature. The chromatograms in Appendix A show the peaks of compounds in the fractions of *C. rosea* TNAU-CR 02. Table 2 indicates the name of the detected compounds, the retention time, peak area percentage, probability, exact mass, molecular formula, NIST similarity in percentage, and reference database compounds. Through GCMS analysis, 18 different bioactive compounds with different antagonistic activity were identified. Further molecular docking work is currently in progress with the identified antimicrobial compounds against the pathogenic virulence protein, succinate dehydrogenase (SDH).

### 2.6. Effects of the Crude Antibiotics of C. rosea TNAU-CR 02 against S. sclerotiorum

The crude antibiotics of the *C. rosea* TNAU-CR 02 isolate were verified for their antifungal activity against *S. sclerotiorum*. The crude antibiotics extracted from *C. rosea* TNAU-CR 02 were poured into wells at the concentration of 25, 50, 75 and 100 ppm at the rate of 10 µL per well per plate. The results revealed that 97.17% inhibition was observed at 100 ppm followed by 75 ppm with an inhibitory percentage of 92.33%. The minimum inhibition was observed at 25 ppm with an inhibitory percentage of 82.17% followed by 87.42%. The results showed that the existence of secondary metabolites in *C. rosea* TNAU-CR 02 inhibited the mycelial growth of the pathogen (Appendix A).

### 2.7. The Effect of C. rosea TNAU-CR 02 Liquid Bioformulation for the Management of Cabbage Head Rot Pathogens under Greenhouse Conditions

The efficacy of a liquid formulation of the *C. rosea* isolate TNAU-CR 02 was evaluated against *S. sclerotiorum* in cabbage plants under greenhouse conditions. The antagonistic fungi and a standard chemical fungicide, tebuconazole + trifloxystrobin (NATIVO) was used to treat for head rot infection. Among the five treatments, the exclusive application of chemical fungicide at 1.5 g/litre on 30, 45 and 60 DAT (days after transplantion) were found to be significantly superior with the lowest disease incidence of 5.9%, compared to 22.8 PDI (percentage disease incidence) against the control. This was followed by foliar application of the *C. rosea* isolate TNAU-CR 02 at 5 mL/litre on 30, 45 and 60 DAT recording the lowest disease incidence of 15.1 PDI. The results clearly indicate that *C. rosea* shows a better percentage disease inhibition compared to other commercially available biocontrol agents (Figure 6, Table 3).

## 3. Discussion

The current study primarily focused on Indian strains of *C. rosea*, a sclerotial mycoparasite of *Sclerotinia sclerotiorum*, its morphological characteristics, molecular analysis, and mode of action against the cabbage head rot pathogen in Tamil Nadu, India. The mycoparasitized sclerotia were collected from Coonoor, Nilgiris district, Tamil Nadu, India. This species produces white to yellowish mycelial growth with multiple branched septate mycelia, hyaline conidiophores, with smooth, lobed to round conidia. The morphological characteristics of our isolates were in accordance with other author’s descriptions [11,12]. The identification of the isolated fungus, *Clonostachys* sp. was further confirmed by molecular characterization using ITS primers. The amplified PCR products produced an amplicon size of about 580 bp corresponding to the ITS region. Similar results were observed by Mohammed et al., (2022) [48] who reported the molecular characterization of two entomopathogenic *C. rosea* using the ITS 1 and 4 universal primers. Dual culture assays were conducted to evaluate the five *C. rosea* isolates against *S. sclerotiorum* [26,49,50]. Although, co-inoculation between pathogens and biocontrol agents showed mutual inhibition [51], prior inoculation of *C. rosea* TNAU-CR 02 showed better antagonistic activity against *S. sclerotiorum* indicating 79.63% inhibition over the control. These results are in accordance with the mycelial inhibition of *S. sclerotiorum* by *C. rosea* BAFC3874 [26]. The SEM images of mycoparasitized sclerotia showed the colonization of *C. rosea* with distorted, shrunken and shrivelled conidia. Similar results were reported by Rodriguez et al. (2011) [26], and Whipps JM (2001) [52] who found that *C. rosea* colonized the sclerotia within 20 days of inoculation, and that sclerotial viability decreased. Only a few fungi can directly parasitize untreated sclerotia [53,54] and as a result, untreated sclerotia can become infected with mycoparasitic fungi. Similar findings were reported by Vinodkumar et al., (2017) [55] who observed the colonization of *T. asperellum* NVTA2 on sclerotial bodies of *S. sclerotiorum* through scanning electron microscopy.

The GC-MS analysis of the bioactive organic compounds secreted in the ethyl acetate fractions of *C. rosea* TNAU-CR 02 showed a total of eighteen bioactive metabolites such as dihydro-3-(2H)-thiophenone, dl-glyceraldehyde dimer, propanedioic acid, etc., with antifungal, antibacterial and antioxidant activities. These compounds with fungal activity may be responsible for the reduction of mycelial growth under in vitro conditions. Rodriguez et al. (2011) [26], reported the antifungal activity of *C. rosea* BAFC3874 isolated from suppressive soils against *S. sclerotiorum* in lettuce and soybean plants. The results of the present study are also in agreement with those of the above authors. Further, the greenhouse experiments explained the efficacy of *C. rosea* TNAU-CR 02 against the notorious cabbage head rot pathogen, showing the lowest disease incidence of 15.1 PDI. Though *C. rosea* TNAU-CR 02 showed the lowest disease incidence compared to other biocontrol agents, a considerably large portion of cabbage head was infected with the pathogen, hence a better formulation must be developed in order to enhance its biocontrol potential against head rot disease. Further, whole genome sequencing, transcriptomic analysis and multi-location trials under various environmental conditions should be conducted to further demonstrate the efficacy of *C. rosea* TNAU-CR 02 to exploit as a better biocontrol agent against *S. sclerotiorum* infection in India.

## 4. Materials and Methods

### 4.1. Sample Collection and Fungal Isolation

The sclerotial bodies of the fungal pathogen, *Sclerotinia sclerotiorum* were collected from the infected head portions of cabbages from Coonoor, Nilgiris district, Tamil Nadu, India (11°24′41.0″ N latitude and 76°42′40.2″ E longitude). The sclerotia that appeared to be infected by mycoparasites presented with white to yellowish mycelial growth. These sclerotia were collected in sterilized plastic bags and sent to the laboratory for isolation and identification. Mycelia were isolated both externally and internally, and the level of fungal infection was visually observed using a stereomicroscope. Similarly, the mycelial characteristics, such as hyphal structure, conidia and condiophores, were observed under a microscope (Olympus BX9—CBH clinical). The collected sclerotia were surface sterilized with 1% sodium hypochlorite followed by washing with sterile distilled water. After sterilization, aseptically treated sclerotia were inoculated in polystyrene dispensable plastic Petri dishes (100 × 15 mm) containing potato dextrose agar (PDA) supplemented with rifampicin at a concentration of 0.1 g dissolved in 1 mL of ethanol/100 mL of potato dextrose agar medium. After 15 days of incubation, at 23 °C under fluorescent light, white to yellowish mycelial colonies appeared on the PDA medium, and the isolated mycoparasite was then observed under a microscope showing morphological qualities including, sporodochia, with numerous condia and conidiospores. These were selected and the cultures were purified further by the single-spore isolation method [56]. The pure culture was maintained at 23 °C and used for further studies. The colony morphology, pigmentation, substrate colour, topography and colony edge were observed and recorded. Similarly, the pathogenic *S. sclerotiorum* isolate TNAU-SS-5 (NCBI accession no MZ379266) was chosen for the present investigations based on its supremacy in our previous experiments. The TNAU-SS-5 isolate was artificially grown on PDA medium and incubated at 25 °C under fluorescent light for 3 days. Subsequently, secondary sclerotia were produced at the border of the plates. These plates were stored at 4 °C until used for antagonism tests and SEM analysis

### 4.2. Morphological and Molecular Characterization of Mycoparasitic Fungal Isolates from Infected Sclerotia

Five isolates of mycoparastic fungi were isolated from the gathered sclerotia of *S. sclerotiorum*. The morphological characteristics of all five isolates were identified based on their dark-yellowish pigmentation on the Petri dish, the presence of orange-coloured sporodochia, conidiophores, conidial colour, and size [11,12]. The 9 mm mycelial plugs of isolated mycoparasitic fungi were positioned in 250 mL Erlenmeyer flasks containing 150 mL potato dextrose broth and incubated at 23 °C for 20 days. Later, the mycelial mats of all the isolates were collected, dried on sterilized paper, and finely crushed using liquid nitrogen with a mortar and pestle. Genomic DNA was obtained from the pure culture of mycoparasitic fungi using cetyl trimethyl ammonium bromide (CTAB) [57]. PCR amplification was carried using ITS 1 (sequence: 5′- TCCGATGGTGAACCTGCGG-3′) and ITS 4 primers (sequence: 5′TCCTCCGCTTATTGATATGC-3′) [58]. The master cycler gradient PCR (Eppendorf) was executed with a 25 µL reaction volume with an initial template DNA step of 95 °C for 10 min, denaturation at 94 °C for 30 s, annealing step at 60 °C for 1 min, extension step at 72 °C for 1 min and the final extension step at 72 °C for 10 min, followed by 35 cycles and held at and 4 °C.

### 4.3. Sequencing and Phylogenetic Analysis

For each PCR reaction, the amplified products were analysed by electrophoresis using a 1% agarose gel to identify estimated base pairs [59]. Then, the PCR products were eluted using a QIAquick^®^ PCR Purification Kit (catalogue numbers. 28104 and 28106) and sequenced by Sanger dideoxy sequencing using the primers ITS 1 and ITS 4 at Chromos Biotech Pvt. Ltd. Bangalore, India. The partial sequence of the 5.8S gene and the flanking internally transcribed spacers (ITS 1 and ITS 4) of the isolated strains were sequenced with BLAST (https://blast.ncbi.nlm.nih.gov/Blast.cgi, accessed on 18 September 2022) against the NCBI GenBank (GenBank; http://www.ncbi.nlm.nih.gov/BLAST/incidence.html, accessed on 18 September 2022) to verify the closest similarity index for molecular taxonomic identification. The isolated strains were submitted to GenBank, and the accession numbers of the sequences were obtained. A phylogenetic tree was constructed using the 5.8 S rRNA gene sequences of the mycoparasitic fungal isolates using the Mega 7.0 software [60] to create a maximum neighbour-joining tree with 1000 boot strap replicates. The nucleotide sequences were aligned using Clustal W for multiple sequence alignments, and a sequence identity matrix was created using bio Edit software (Version 7.0.4.1).

### 4.4. Dual Culture Assays

The efficacy of mycoparasitic fungal isolates obtained from the sclerotial bodies of cabbage were tested in vitro using the dual culture technique against *S. sclerotiorum* TNAU-SS-5 (NCBI accession no MZ379266) [61] obtained from the culture collection centre, Department of Plant Pathology, Tamil Nadu Agricultural University, Coimbatore, Tamil Nadu, India. Two evaluations were made: a 9 mm diameter disc of actively growing mycelia of a 20-day-old mycoparasitic fungal culture was inoculated for 7 days before pathogen inoculation or at the same time of the pathogen inoculation. In both evaluations, a single, 9 mm diameter mycelial disc of 7-day-old culture of *S. sclerotiorum* was excised from the actively growing edge and placed on the opposite side of the PDA dishes [61]. Three replications were kept, and the medium with just pathogen inoculation was kept as a control. The plates were then incubated at 23 °C for 20 days, with 10 days of darkness and 10 days of light. During this period, the plates were examined for the formation of inhibition zones between the growth of mycoparasites and *S. sclerotiorum*. The isolates with the highest inhibition were chosen and used in future experiments.

### 4.5. Preparation of Conidiospore Suspension of Effective Mycoparasitic Fungal Isolates

After examining the conidiophore and conidial production of the effective *C. rosea*, TNAU-CR 02 isolate, the spore suspension was prepared by harvesting the conidia from the 3-week-old colonies grown on PDA at 23 °C. Conidia of *C. rosea*, TNAU-CR 02 were washed off the Petri dish by gently rubbing the sporodochia and mycelial colonies with a sterile brush and 5 mL sterile distilled water containing 1% fructose to release spores [62]. The final concentration of the undiluted spore suspension was calculated using a haemocytometer and adjusted to 1 × 10^7^ spore mL^−1^. This concentrated spore suspension was sprayed over healthy sclerotia to assess the mycoparasitic activity of *C. rosea* TNAU-CR 02.

### 4.6. Scanning Electron Microscopy Analysis

The stockpiled sclerotia from the TNAU-SS-5 isolates were collected. Ten sclerotia were randomly selected and artificially placed on sterilized soil in Petri dishes. Then, the sclerotia were spray-inoculated with 5 mL of the prepared conidial suspension of *C. rosea*, TNAU-CR 02 (1 × 10^7^ spore mL^−1^). The inoculated plates were examined at different intervals for their mycoparasitic activity, 20 days after spray inoculation, three sclerotia were randomly selected. These selected sclerotia were bisected using a sterilized razor blade and half of the sclerotia were directly fixed to copper SEM stubs using double-sided sticky tape. Then, the fixed sclerotial samples were super-coated with a gold alloy utilizing a EMITECH ion sputter coater to a thickness of 300 N reducing injury to the samples during electron imaging. The pictures were captured using a FAI QUANTA 250 Model SEM at 15 KV. During SEM imaging, the mechanism and mode of action of *C. rosea*, TNAU-CR 02 over the sclerotia of *S. sclerotiorum* was documented.

### 4.7. GC-MS Analysis of the Volatile Organic Compounds and Non-Volatile Organic Compounds Extracted from the Ethyl Acetate Fractions of C. rosea TNAU-CR 02

The crude antibiotics of the effective isolate, *C. rosea* TNAU-CR 02, were examined through GC-MS (GC Clarus 500 Perkin Elmer) analysis. In brief, among the five *C. rosea* fungal isolates, TNAU-CR 02 showed the maximum mycelial inhibition in the dual culture assays and thus was selected to identify the volatile and non-volatile organic compounds responsible for the suppression of *S. sclerotiorum* through GC-MS analysis [63]. The biochemical compounds (BCs) produced by *C. rosea* TNAU-CR 02 in the PDA medium from the inhibition zone were separated and the fractions were prepared by scraping using a sterile scalpel. 150 mg of excised agar medium, 500 μL of HPLC-grade acetonitrile:water (1:1; *v*:*v*) were mixed. Then, the mixture was sonicated (Bandelin Sonoplus HD 2070) twice for 30 s at 30% power. Subsequently, the samples were mixed (vortex), centrifuged and sorted to eliminate any particles. The extracts were finally concentrated to 150 mL by evaporation under reduced pressure using a vacuum flask evaporator (Rotrva Equitron Make) and the remaining solvent was air-dried in the sterilized Petri dish. After removing the eluent, the final products of the metabolites were dissolved in 1 mL of HPLC-grade methanol. The resulting mixtures were filtered with a 0.25 μm polyvinylidene difluoride membrane syringe-like filter and extracted using a sodium sulfate cartridge to remove water and then concentrated to 1 mL with a turbovap at 55 °C with nitrogen before GC-MS analysis [64].

### 4.8. Bioassays of the Antifungal Activities of C. rosea against S. sclerotiorum

The inhibitory response of the crude antibiotics of *C. rosea* TNAU-CR 02 on the mycelial growth of *S. sclerotiorum* was analysed by agar well diffusion in vitro [65]. PDA medium was poured into the Petri dish and after solidification, with the support of cork, a 9 mm diameter agar well was made in the medium on all the four sides of the Petri dish, 1 cm from the rim. Then, the vigorously growing 9 mm mycelial disc of *S. sclerotiorum* TNAU-SS-5 was placed in the middle of the Petri dish. The extracted crude antibiotic metabolites of *C. rosea* TNAU-CR 02 were poured into each well at a rate of 10 µL per well per plate with the same concentration in each of four wells per plate. Four different concentrations were used *viz.*, 25, 50, 75, 100 ppm and the plates were incubated for 3 days at 20 °C. The inhibitory percentage of pathogen due to the presence of metabolites from *C. rosea* TNAU-CR 02 was calculated. Each dose was replicated thrice and HPLC-grade methanol was kept as a control to estimate the effect of the crude metabolites.

### 4.9. Development of a Liquid Formulation of the C. rosea TNAU-CR 02 Isolate

A liquid bioformulation was developed by inoculating the 9 mm mycelial disc of *C. rosea* TNAU-CR 02 in yeast molasses broth (YEM). Biocontrol-inoculated broth was incubated at 20 ± 2 °C for 15 days in an orbital shaker at 130 rpm. The mycelial mat along with the broth was blended and the mixture was centrifuged at 5000× *g* rpm for 2 min and the supernatant spore suspension was collected. The concentration of conidiospores in the collected spore suspension was determined using a bio-spectrophotometer with an optical density (OD) of 0.650. The suspended conidiospores were blended along with 10 mL of 2% glycerol and 1% of polyvinyl alcohol (synthetic hydrophilic polymer). The concentration of the resultant conidiospore counts was determined using a haemocytometer and the resultant spore suspension was adjusted to 2 × 10^8^ cfu/mL.

### 4.10. Fungal and Bacterial Isolates

The efficacy of a liquid formulation of *C. rosea* TNAU-CR 02 was tested in comparison to other commercially available biocontrol agents against *S. sclerotiorum* in cabbage plants under greenhouse conditions. *T. asperellum* TRI 15 (KX533985), and the bacterial antagonist *B. subtilis* (MG241251) were obtained from the culture collection centre, Department of Plant Pathology, Tamil Nadu Agricultural University, Coimbatore along with a standard chemical fungicide, Tebuconazole + Trifloxystrobin, to assess the effectiveness of the mycoparasitic fungal isolate, *C. rosea* TNAU-CR 02.

### 4.11. Pathogen Mass Multiplication

For assessing the greenhouse experiments, soil inoculated with *S. sclerotiorum* was prepared in sand–maize medium (Riker and Riker, 1936). Sand and ground maize seeds were combined 19:1; saturated with 100 mL of water per 500 g and covered with polypropylene bags. The prepared medium was autoclaved twice on alternate days. A 9 mm mycelial disc of *S. sclerotiorum* was inoculated into the sand maize medium and incubated at room temperature for 15 days. Earthen pots 30 cm in diameter were filled with 5 kg of pot mixture (red soil:sand:FYM at 1:1:1 *w*/*w*/*w*). The pot mixture was sterilized twice on two successive days. Pots containing autoclaved soil were inoculated with sand–maize inoculum at 50 g/kg soil and left for 5 days for the establishment of the inoculum.

### 4.12. The Efficacy of the Biocontrol Agents on the Severity of Cabbage Head Rot Pathogens under Greenhouse Conditions

The experiment was laid out to study the effect of the potential biocontrol agents on the disease incidence of cabbage head rot under greenhouse conditions at the Department of Plant Pathology, Tamil Nadu Agricultural University, Coimbatore, Tamil Nadu, India. Susceptible cabbage seedlings (Hybrid giant) were raised in the department nursery and the 45-day-old seedlings were used in the greenhouse experiments. The experiment consisted of six treatments in a completely randomized block design *viz.*, T1—foliar application of *C. rosea* TNAU-CR 02 at 5 mL/litre on 30, 45 and 60 DAT; T2—foliar application of *Trichoderma asperellum* (TRI 15) at 5 mL/litre on 30, 45 and 60 DAT; T3—foliar application of *Bacillus subtilis* (Bbv 57) at 5 mL/litre on 30, 45 and 60 DAT; T4—tebuconazole + trifloxystrobin at 1.5 g/litre on 30, 45 and 60 DAT; T5—inoculated control; T6—uninoculated control. All the treatments were replicated three times with 20 plants per replication, maintained along with the standard chemical fungicides and untreated controls. 45-day-old susceptible cabbage seedlings (hybrid giant) were dipped in the corresponding treatments, transplanted into prepared pot mixtures and challenged with *S. sclerotiorum*. Concurrently, the untreated control was treated with water. The treated plantlets were maintained in a controlled atmosphere-conditioned greenhouse with a temperature of 22/20 °C day/night with a minimum photoperiod of 16 h of light and 8 h of darkness. Disease incidence was assessed after 30, 45 and 60 DAT showing evident symptoms of water-soaked lesions spreading quickly with the presence of fluffy mycelial growth. The infestation was assessed using the following formula
Percent Disease Incidence = No. of infected plants/Total No. of plants × 100(1)

### 4.13. Statistical Analysis

All determinations and extractions were performed in triplicate, and the results are expressed as mean ± standard deviation (n = 5). Statistical analyses were performed for dual culture assays and crude antibiotic metabolite assays using XLSTAT software version 19 (XLSTAT, New York, NY, USA). The values in the parentheses for dual culture assays and biocontrol efficacy under greenhouse conditions represent arcsine-transformed values.

## 5. Conclusions

The present study provides the first report of *C. rosea* as an mycoparasitic fungus of the cabbage head rot pathogen, *S. sclerotiorum*. The investigated fungus was not reported to colonize the sclerotia. This discovery expands our understanding of the biology and dissemination of *C. rosea*. The role of *C. rosea* as a mycoparasitic fungus on sclerotia needs additional investigation, to clarify the molecular mechanisms behind the sclerotial fungal infection. The observations from the SEM analysis clearly reveal an antagonistic activity of *C. rosea* against the pathogen with the formation of appresoria, and the production of numerous conidia over the outer cortex layer resulting in the disintegration of *S. sclerotiorum* sclerotia. By inspecting the chemical compounds in the ethyl acetate fractions of the *C. rosea* TNAU-CR 02 isolate, it additionally paves a way for understanding its multivarious activity against fungal growth. Moreover, foliar application of TNAU-CR 02 at 5 mL/litre on 30, 45 and 60 days after transplantation showed the lowest disease incidence of 15.1 PDI. To increase the biocontrol capability of *C. rosea* TNAU-CR 02 against the pathogen, a stronger formulation must be developed to enable its marketing. Clearer knowledge of the complex exchanges between the pathogen and the biocontrol agent will provide a way for the exploitation of *C. rosea* against cabbage head rot pathogens.

## Figures and Tables

**Figure 1 plants-12-00199-f001:**
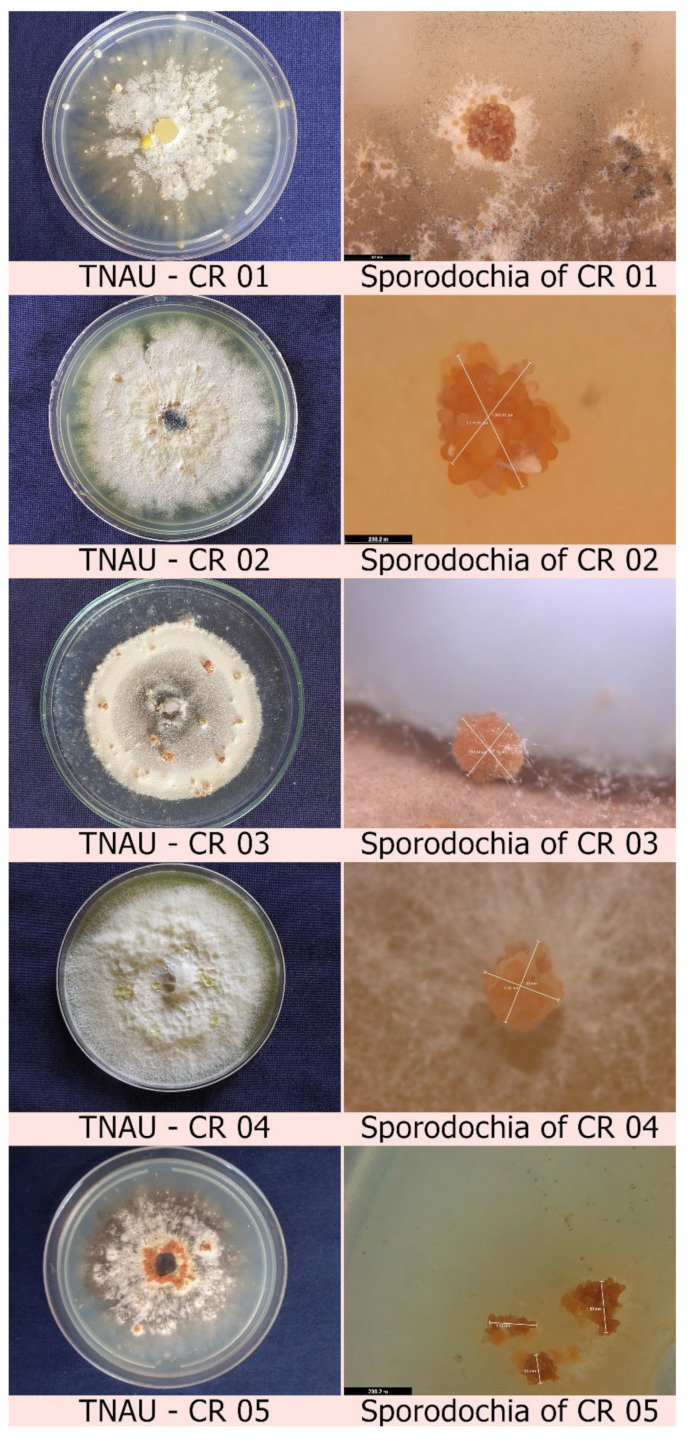
Colony morphology and sporodochial characteristics of all five *C. rosea* isolates.

**Figure 2 plants-12-00199-f002:**
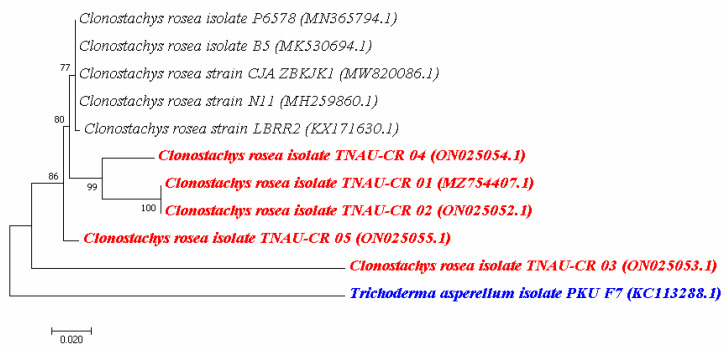
Phylogenetic analysis of *C. rosea* isolates. (**Red color**—Indicates different *C. rosea* isolates; **Blue color**—Indicates out group).

**Figure 3 plants-12-00199-f003:**
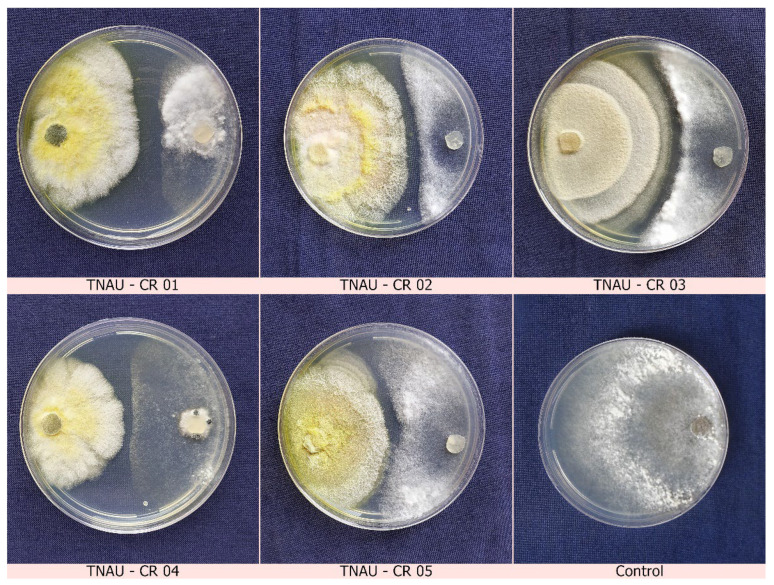
Dual culture assays of *C. rosea* against *S. sclerotiorum* twenty days after inoculation (**left side**—biocontrol agent, *C. rosea*; **right side**—pathogen, *S. sclerotiorum*).

**Figure 4 plants-12-00199-f004:**
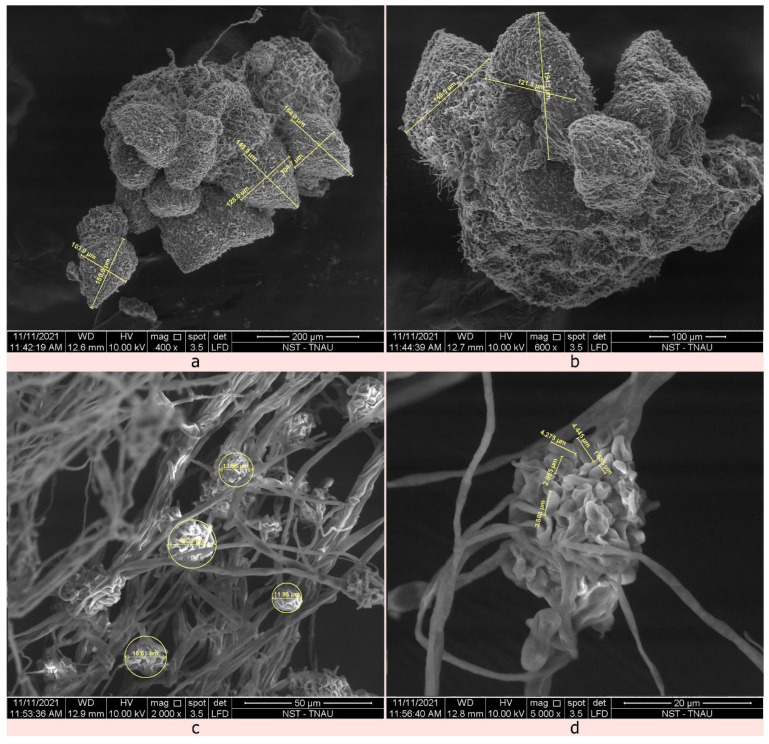
Structural components of *C. rosea* TNAU-CR 02 isolate. (**a**,**b**) Irregular-shaped sporodochia (**a**) 400× magnification; 200 µm and (**b**) 500× magnification; 100 µm). (**c**,**d**) Conidiophore with conidia (**c**) 2000× magnification; 50 µm and (**d**) 5000× magnification; 20 µm).

**Figure 5 plants-12-00199-f005:**
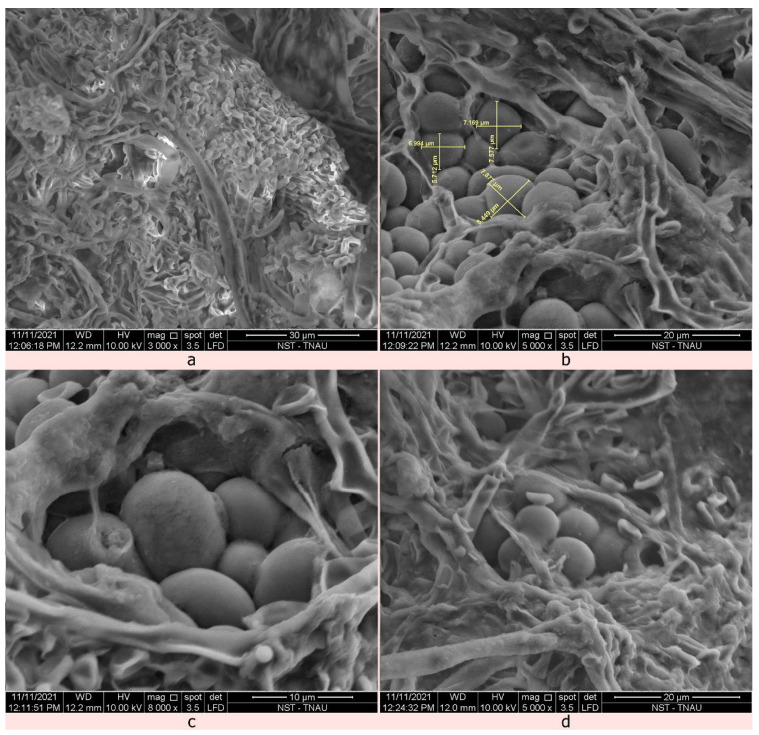
SEM images of *C. rosea* interaction over the sclerotia. (**a**) Presence of numerous conidiospores surrounding the outer sclerotial wall (3000× magnification; 30 µm). (**b**) A closer view of the *C. rosea* conidia wall (5000× magnification; 20 µm). (**c**) The development of appressoria-like spore attachment structures (8000× magnification; 10 µm). (**d**) Spore contact at the cortex layer of the sclerotia (5000× magnification; 20 µm).

**Figure 6 plants-12-00199-f006:**
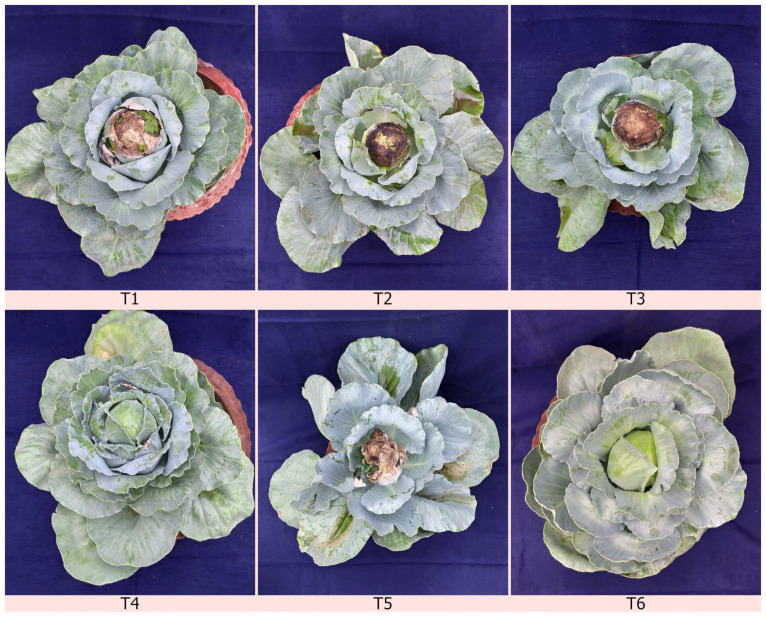
The effect on biocontrol agents on the severity of cabbage head rot under greenhouse conditions. (**T1**) Foliar application of *C. rosea* TNAU-CR 02 at 5 mL/litre on 30, 45 and 60 DAT. (**T2**) Foliar application of *Trichoderma asperellum* (TRI 15) at 5 mL/litre on 30, 45 and 60 DAT. (**T3**) Foliar application of *Bacillus subtilis* (Bbv 57) at 5 mL/litre on 30, 45 and 60 DAT. (**T4**) Tebuconazole + trifloxystrobin application at 1.5 g/litre on 30, 45 and 60 DAT. (**T5**) Inoculated control. (**T6**) Uninoculated healthy control.

**Table 1 plants-12-00199-t001:** Dual culture assays of *C. rosea* against *S. sclerotiorum* twenty days after inoculation.

Sl.No.	Isolate	Percent Inhibition over Control *
1	TNAU-CR 01	70.37 ^c^ (57.02)
2	TNAU-CR 02	79.63 ^a^ (63.17)
3	TNAU-CR 03	74.07 ^b^ (59.39)
4	TNAU-CR 04	52.22 ^e^ (46.27)
5	TNAU-CR 05	58.14 ^d^ (49.68)

* Values are the means of three replicates. Means in a column followed by same superscript letters are on par with each other according to LSD. Values in parentheses represent arcsine transformation.

**Table 2 plants-12-00199-t002:** Metabolite fingerprinting of the volatile/non-volatile bioactive compounds extracted from the ethyl acetate fractions of *C. rosea* TNAU-CR 02.

Sl.NO	Compound	Retention Time	Peak Area Percentage	Probability	Activity	References
1	Dihydro-3-(2H)-thiophenone	3.123	0.277	11.2	Antifungal activity	[30]
2	dl-Glyceraldehyde dimer	3.324	1.774	59.3	Insecticidal property	[31]
3	Propanedioic acid, phenyl-	8.301	0.553	19.5	Insecticidal property	[32]
4	Butanamide, 2-hydroxy-N,2,3,3-tetramethyl	10.827	0.155	9.8	Antiviral activity	[33]
5	à-Methyl 4-O-methyl-D-mannoside	11.237	0.392	29.9	Antifungal activity	[34]
6	4-Methyl(trimethylene)silyloxyoctane	12.187	0.188	15.8	Antioxidant and antimicrobial activity	[35]
7	1-Tetradecanamine, N, N-dimethyl-	17.034	0.406	60.2	Antifungal activity	[36]
8	Pilocarpine	20.67	0.149	37.6	Mutagenicity, cytotoxicity, and antimicrobial activity	[37]
9	Hexadecanoic acid, 2-hydroxy-1-(hydroxymethyl)ethyl ester	21.281	1.356	52.3	Antifungal activity	[38]
10	Pentadecanoic acid, 13-methyl-, methyl ester	21.541	0.165	24.5	Anti-bacterial activity	[39]
11	9-Hexadecenoic acid	21.751	0.49	9.9	Antifungal activity	[40]
12	Octadecane, 3-ethyl-5-(2-ethylbutyl)-	21.936	0.169	10.6	Nematicidal property	[41]
13	Erucic acid	24.047	1.088	11.2	Antifungal activity	[42]
14	8,11-Octadecadienoic acid, methyl ester	24.697	0.178	9.1	Antifungal activity	[43]
15	9-Octadecenoic acid, methyl ester, (E)-	24.822	0.289	7	Antifungal activity	[44]
16	13-Docosenamide, (Z)-	26.358	4.829	72.4	Antifungal activity	[45]
17	cis-11-Eicosenamide	28.354	0.138	4.5	Antibacterial activity	[46]
18	17-Pentatriacontene	28.554	0.262	10.9	Antifungal activity	[47]

**Table 3 plants-12-00199-t003:** Efficacy of biocontrol agents on the severity of cabbage head rot pathogens under greenhouse conditions.

T. No.	Type of Treatments	Disease Incidence (Per Cent Disease)		
		30 DAT	45 DAT	60 DAT
T1	Foliar application of *C. rosea* isolate TNAU-CR 02 at 5 mL/litre on 30, 45 and 60 DAT	5.9 ^b^ (14.05)	8.3 ^b^ (16.73)	15.1 ^b^ (22.86)
T2	Foliar application of *Trichoderma asperellum* (TRI 15) at 5 mL/litre on 30, 45 and 60 DAT	6.4 ^c^ (14.64)	9.1 ^c^ (17.55)	17.9 ^c^ (25.02)
T3	Foliar application of *Bacillus subtilis* (Bbv 57) at 5 mL/litre on 30, 45 and 60 DAT	6.9 ^bc^ (15.22)	9.7 ^b^ (18.14)	17.1 ^b^ (24.41)
T4	Tebuconazole + trifloxystrobin at 1.5 g/litre on 30, 45 and 60 DAT	0.0 ^a^ (2.86)	2.2 ^a^ (8.52)	5.9 ^a^ (14.05)
T5	Inoculated control	9.5 ^d^ (17.95)	16.5 ^d^ (23.96)	22.8 ^d^ (28.52)

Values are the means of five replications. Means in a column followed by the same superscript letters are not significantly different according to DMRT at *p* ≤ 0.05. Values in parentheses represent arcsine transformations.

## Data Availability

The dataset supporting the conclusions of this article are included within the article and its Appendix A.

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
