# Peer review of "First Report of Clonostachys rosea as a Mycoparasite on Sclerotinia sclerotiorum Causing Head Rot of Cabbage in India"

_plants, 2023, doi:10.3390/plants12010199_

Round 1

Reviewer 1 Report

Dear editor and authors,

Thanks for the opportunity to review this interesting manuscript. I have, however, a few comments that need some attention.

Abstract

Line 16: A total of five C. rosea were isolated consider revising to « A total of five isolates….. were found infected ….»

Introduction

Line 30: host range, worldwide infects consider revising to …. «worldwide. It infects….»

Line 31-33: Consider revising to «The pathogen is able to survive in soil for longer periods of time as a type of resting structure called a sclerotium [2, 3] because of its hard rind melanized layer that can withstand chemical and physical adverse environmental conditions [4.»

Line 42-45: consider revising to «While the pathogen is extremely challenging to manage because of its survivability in the resting structure, a variety of management methods are being utilized to decrease their survival and abundance.»

Results

Line 115: not «similar» but «close» species

Line 199: indicate the clusters on the figure 2

Figure 2: As far as I can see two strains are identical, while you state that isolates differ from each other

Line 128-129: consider revising to «Incubating C. rosea and S. sclerotiorum on PDA dishes simultaneously shows colonies growing towards one another without evidence of inhibition zones by…»

Line 131-132: consider revising to «When C. rosea strains mixed with S. sclerotiorum colonies, hyperparasitism was observed in abundance on the interaction site about 10 days into the interaction. The mycelium started infecting and covering the sclerotia, forming small yellow colonies over the sclerotia to reveal the mycoparasitism.

Line 133-134: consider revising to «inoculation, C. rosea covers the sclerotial bodies of pathogen, killing and mycoparasitizing the resting stage.

Line 136: invades

Line 139: greatestthe highest

Line 190: shows show

Line 215: inhibited the mycelial

Line 223-224: as it recorded the lowest disease incidence of 5.9 percent, compared to 22.8 PDI for the control.

Line 248: ……India. This species produces …..

Line 257-258: consider revising to «Although, co-inoculation between pathogens and biocontrol agents shows mutual inhibition [51], prior inoculation of C. rosea TNAU-CR 02 showed antagonistic activity of

Line 261-262: The SEM image of mycoparasitized sclerotia shows the colonization of C. rosea with distorted, shrunken, and shriveled conidia.

Line 264- 265: who found that C. rosea colonized sclerotia within 20 days of inoculation, and that sclerotial viability decreased.

Line 277-278: The results of the present study were also in agreement with those of the above authors.

Line 282: as a better

Line 288-290: The Sclerotia that appeared to be mycoparasites with white to yellowish mycelial growth were collected in sterilized plastic bags and sent to the laboratory for isolation and identification.

Conclusions

The following sentences are more about MM, and not conclusion

All determinations and extractions were performed in triplicates, and the results were expressed as mean ± standard deviation (n = 5). Statistical analysis was performed for dual culture assay and crude antibiotic metabolite assay using XLSTAT software version 19 (XLSTAT, New York, USA).

Reviewer 2 Report

To the authors, 

Major comment: 

Foliar application of C. rosea seems somehow strange as it was isolated from the soil and on sclerotia. As such, could soil application be a better way to control the disease in a early stage? 

Also, I'm missing a conclusion? See PDF. 

And for the metabolite fingerprinting: It is not clear for me what the added value is of this test as you did it only for TNAU-CR02 and not for the other isolates. As such, it is hard to compare the isolates with each other and  understand why 02 is better than the rest. Moreover, as you stated in M&M; you wanted to unveil the bioactive metabolites responsible for suspression of Sclerotinia but whit this data it is not clear to me which compound(s) is responsible for this suspression. 

As last major comment; in the bio-assay, although isolate 2 had the lowest disease incidence for the BCO's, the pictures indicate still quiet a large affected area in the head of the plant. I think it's is still not marketable? As such, a multiple application is needed or a better formulation is needed. You should headlight this in your discussion and conclusion.

Yours sincerely 

Reviewer 3 Report

I find the research valuable. However, several important points need clarification. It is not clear how the health evaluation was carried out in the greenhouse experiment? How many plants were tested on? Relatively few new literature items were used in the preparation of the manuscript.  The discussion of the results is very brief. All results should be discussed with the available literature. The conclusion of the work is lacking. Overall, the manuscript was prepared with little care therefore it needs major revisions.

Specific comments:

1. The title is not appropriate. Clonostachys rosea has broad antagonistic properties against Sclerotinia sclerotiorum as documented in the paper.

2. The abstract should include the most important results of the study.

3. Keywords should not duplicate the iznoramptions contained in the title of the paper.

4. L87. 2.1 Isolation of Clonostachys rosea from the Mycoparasitized Sclerotia. From which Sclerotia were strains of Clonostachys rosea isolated? Remove Mycoparasitized from the title.

5. L104. Molecular Characterization of Clonostachys rosea isolatetes.

6. L123. Phylogenetic analysis of C. rosea isolates- correct title. Fig 2. Not readable, should be improved.

7. L133. Twenty days after inoculation- was photographic documentation done?

8. L145. Figure 3- Photo shows Dual Culture Assays- current title is not correct. On which day of the experience were photographs taken?

9. L146. Table 1--shows the results of the Dual Culture Assays--the title is not correct. On which day of the experience?

10. L150. 2.4 Identification of the Mycoparasitic Growth of C. rosea.

11. No documentation cited. On what basis were the observations made? Combine sections 2.4 and 2.5.

12. L228-231 Fig. 6. Illegible captions for figures T1-T6 (font too small).

No explanation of what 30, 45 and 60 DAT means. How many days after the experiment was set up? The study included only biocontrol agents? Title is not appropriate because it does not cover all aspects of the experiment.

13. L287. Provide GPS coordinates.

14. L322. 4oC - enter correct sign.

15. L323. What sequencing method was used? On what equipment was the research performed? In what laboratory?

16. L346. 23oC for 20 days- -enter correct sign.

17. L 435. 4.11. Efficacy of Biocontrol Agents on the Severity of Cabbage Head Rot Pathogen Under Glass House Conditions- Describe in detail how the infestation was assessed?

18. L449. Conclusions does not include a summary. Be sure to complete them.

19. L450-454- Statistical analysis? Did it use transformation? All data on the use of mathematical methods should be included here.

Round 2

Reviewer 2 Report

To tha authors, 

Thanks to implement the suggested recommendations. 

With kind regards.

Reviewer 3 Report

The manuscript has been revised and I therefore recommend its acceptance for publication in PLANTS MDPI journal.